# Tick-Borne Encephalitis Virus Vaccination among Tourists in a High-Prevalence Area (Italy, 2023): A Cross-Sectional Study

**DOI:** 10.3390/tropicalmed8110491

**Published:** 2023-11-02

**Authors:** Matteo Riccò, Silvia Corrado, Federico Marchesi, Marco Bottazzoli

**Affiliations:** 1Occupational Health and Safety Service on the Workplace/Servizio di Prevenzione e Sicurezza Ambienti di Lavoro (SPSAL), Department of Public Health, AUSL–IRCCS di Reggio Emilia, 42122 Reggio Emilia, Italy; 2ASST Rhodense, Dipartimento della donna e Area Materno-Infantile, UOC Pediatria, 20024 Garbagnate Milanese, Italy; scorrado@asst-rhodense.it; 3Department of Medicine and Surgery, University of Parma, 43126 Parma, Italy; federico.marchesi@unipr.it; 4Department of Otorhinolaryngology, APSS Trento, 31223 Trento, Italy; marco.bottazzoli@apss.tn.it

**Keywords:** tick-borne diseases, vaccines, health knowledge, attitudes, practices, encephalitis, tick-borne encephalitis virus, travel medicine

## Abstract

Tick-borne encephalitis (TBE) represents a potential health threat for tourists in high-risk areas, including the Dolomite Mountains in northeastern Italy. The present questionnaire-based survey was, therefore, designed in order to assess knowledge, attitudes, and preventive practices (KAP) in a convenience sample of Italian tourists visiting the Dolomite Mountains, who were recruited through online discussion groups. A total of 942 participants (39.2% males, with 60.2% aged under 50) filled in the anonymous survey from 28 March 2023 to 20 June 2023. Overall, 24.1% of participants were vaccinated against TBE; 13.8% claimed to have previously had tick bites, but no cases of TBE were reported. The general understanding of TBE was relatively low; while 79.9% of participants acknowledged TBE as a potentially severe disease, its occurrence was acknowledged as high/rather high or very high in the Dolomites area by only 51.6% of respondents. Factors associated with the TBE vaccine were assessed by the calculation of adjusted odds ratios (aOR) and 95% confidence intervals through a logistic regression analysis model. Living in areas considered at high risk for TBE (aOR 3.010, 95%CI 2.062–4.394), better knowledge on tick-borne disorders (aOR 1.515, 95%CI 1.071–2.142), high risk perception regarding tick-borne infections (aOR 2.566, 95%CI 1.806–3.646), a favorable attitude toward vaccinations (aOR 3.824, 95%CI 1.774–8.224), and a tick bite(s) in a previous season (aOR 5.479, 95%CI 3.582–8.382) were characterized as being positively associated with TBE vaccination uptake. Conversely, being <50 years old (aOR 0.646, 95%CI, 0.458–0.913) and with a higher risk perception regarding the TBE vaccine (aOR 0.541, 95%CI 0.379–0.772) were identified as the main barriers to vaccination. In summary, tourists to the high-risk area of the Dolomites largely underestimate the potential occurrence of TBE. Even though the uptake of the TBE vaccine in this research was in line with European data, public health communication on TBE is required in order to improve acceptance of this effective preventive option.

## 1. Introduction

Tick-borne encephalitis (TBE) is an acute clinical syndrome caused by the TBE virus (TBEV) [1,2,3,4], an RNA virus belonging to the family of *Flaviviridae*, genus Flavivirus [1,5,6], which targets the central nervous system (CNS) [6,7,8,9]. TBEV is currently classified into five subtypes, but nearly all the cases reported from the European Union/European Economic Area (EU/EEA) are associated with the TBEV-European (Eu) subtype, transmitted by the hard tick *Ixodes ricinus* [1,6]. TBEV-Eu usually causes infections that follow a biphasic course [3,10]. In the first phase, which occurs in 1/3 of cases after an incubation of around 7 days [1,11,12], viraemia is characterized by non-specific, influenza-like symptoms (e.g., fever, fatigue, headache, myalgia, and nausea), usually lasting from 2 to 10 days. In the second phase, which develops in 20 to 30% of cases after an interval lasting from 1 to 33 days, the pathogen enters the CNS and causes severe complications such as meningitis, meningoencephalitis, and myelitis, and even death [1,13,14]. The case-fatality ratio (CFR) for TBEV-Eu infections has been estimated at 0.9% [15], rarely exceeding 1.5% [16], but around 50% of the total number of patients develop a postencephalitic syndrome characterized by cognitive and neurological features, such as paralysis of the cranial nerves and hearing loss [15,17,18].

According to the European Centre for Disease Prevention and Control (ECDC), between 2015 and 2021, a total of 20,237 TBE diagnoses have been reported from EU/EEA member states [3,10]. Age-adjusted notification rates have increased from 0.39 to 0.88 cases per 100,000 people, and the underlying causes have been tentatively identified in relation to climate change (which increases the distribution of *Ixodes* and competent TBEV hosts) and behavioral factors [19,20]. Inappropriate levels of awareness of TBE among population groups at higher risk for tick bites would in turn impair their use of preventive options [21,22], ranging from behavioral strategies (e.g., avoiding off-trail activities, using appropriate personal protective equipment [PPE], etc.), to the uptake of TBE vaccines [13,17,23,24]. In fact, two TBE vaccines (effective in 96% to 99% of cases) are commercially available in Western Europe, i.e., FSME-IMMUN^®^ (Pfizer SRL, Latina, Italy; which is the only vaccine licensed in Italy) and ENCEPUR^®^ (Bavarian Nordic, Hellerup, Danemark) [16], and both immunizations can be used in adults and children ≥ 1 year old who live in high-risk TBE areas or who frequently visit forests and grasslands in high-risk areas [16,24].

With an age-adjusted notification rate not exceeding 0.1 cases per 100,000 head of population [3], Italy is considered at low risk for TBE [16,25,26,27]. However, 94.1% of 203 Italian TBE cases reported from 2017 to 2023 were clustered within the subalpine autonomous provinces of Trento and Bolzano, and the northeastern regions of Veneto and Friuli-Venezia-Giulia (collectively known as the “Triveneto”) [22,25] (Figure A1 in Appendix B). The corresponding pooled notification rate for Triveneto has been estimated at 0.35 cases per 100,000 head of population (95% confidence interval [95%CI] 0.28–0.42) for the timeframe 2017–2020 [3,28,29,30]. Available estimates are, therefore, substantially lower than those from the nearby countries of Austria (range 2017–2021: 1.15–2.74 per 100,000) and Slovenia (range 2017–2021: 2.38–8.75 per 100,000), but some foci within the Triveneto have been characterized by notification rates exceeding 5 cases per 100,000 persons per year [21,25], which is the cut-off suggested by the World Health Organization (WHO) for high-risk areas where the TBE vaccine should be actively offered as a preventive option [16,18,27]. Some high-risk areas have been identified within the Dolomite Mountains [25], a popular holiday destination [21]. The Dolomites are characterized by large forests that provide an appropriate habitat for both *Ixodes* vector and suitable TBEV hosts (i.e., migratory birds and ungulates) [31], while the high number of mountain tourists and workers involved in outdoor activities (e.g., hunting, fishing, camping, collecting mushrooms and berries, forestry, farming, and even military training) increases the opportunities for human infections [21,22], in particular from April to November [21], when the ticks’ activity reaches its seasonal peak [13,18,23,24,32]. As very little is known about the awareness of TBE among tourists to the Dolomites area [31], the present study was designed in order to ascertain the following endpoints: (1) tourists’ awareness of the TBE and tick-borne infections; (2) their level of self-perceived risk for TBEV infection; (3) acceptance of the TBE vaccine.

Our results can, therefore, lead to the appropriate identification of key topics for the proper design of tailored prevention and control programs, not only from a travel medicine point of view but also for the general population.

## 2. Materials and Methods

### 2.1. Study Design and Sample Size

The present study was designed as a cross-sectional questionnaire-based survey (see the STROBE Checklist in Appendix A), with a targeted convenience sample of around 1000 respondents. The minimum sample size (N) was calculated by assuming the annual incidence of tick bites to be 30.2% (a figure that we identified in a previous survey [22]), a Type I error of 5% (0.05), and a power of 95%:N = 1.96^2^ × 0.302 × (1 − 0.32)/0.05^2^ = 3.8416 × 0.302 × 0.698/0.0025 = 324(1)

The questionnaire was delivered between 28 March 2023 and 20 June 2023 across 45 Facebook groups discussing tourism in the Dolomites. After the removal of shared subscriptions, a total of 20,444 unique members were potentially reached. Before posting the study invitation, the chief researcher (M.R.) contacted the responsible administrators and asked for their authorization to share an invitation link to the questionnaire. The link, in turn, led the potential participants to the informed consent form (Appendix A) detailing the aims of the study and reassuring all participants about the anonymous design of the questionnaire, guaranteeing the confidentiality of all retrieved information and that all data would only be stored for the time needed for collective analyses. Only individuals who formally agreed to participate by answering a specifically designed dichotomous item could access the first page of the web survey (Google Forms; Google LLC; Menlo Park, CA, USA). No monetary or other compensation was offered to the participants. 

### 2.2. Inclusion Criteria

The inclusion criteria were: (1)being aged 18 years or older;(2)having any previous knowledge of TBE;(3)having traveled to the Dolomite Mountains as a tourist in the 5 years before the inception of the questionnaire.

All the aforementioned criteria were assessed by three dichotomous (yes vs. no) questions, and only participants fulfilling the three criteria were able to access the subsequent sections of the questionnaire. 

### 2.3. Instruments

The questionnaire was originally developed for a similar intervention among agricultural workers from the Autonomous Province of Trento and has been described elsewhere [21,22]. Regarding its content, retrieved personal data were deliberately restricted to general information that did not allow for the identification of the respondent. Moreover, an IP address, email address, or personal information unnecessary to the survey was neither requested nor retrieved. The questionnaire included the following areas of inquiry (Appendix A).

#### 2.3.1. Demographic Characteristics of the Participants

Data were gathered regarding age (by decennial groups); gender; any occupational background in healthcare settings and agricultural settings; the Italian Region where the respondents mainly worked and lived; their highest educational achievement; and having or not having any pet in their household.

#### 2.3.2. Knowledge Test

According to the original design of Zingg and Siegrist [33], participants received a previously validated knowledge test, containing a total of 12 true–false statements on TBE and tick-borne diseases, the design of which has been described elsewhere [21]. Each correctly answered question added +1 to a sum score (general knowledge score, GKS) while each wrong, missing, or “don’t know” answer added 0 to the sum score (potential range: 0 to 12). Participants were then asked to rate, through a 5-point Likert scale (from 1, “totally disagree”, to 5, “totally agree”), a series of acute (*n* = 10) and chronic (*n* = 9) signs and symptoms possibly related to previous tick bites. Answers were then dichotomized as somewhat agree (score 4 and 5) vs. somewhat disagree (score 1 to 3), and a sum score (symptom knowledge score, SKS; potential range, 0 to 19) was then calculated similarly to the GKS; when the participant agreed on a sign/symptom that was actually associated with TBE, +1 was added to the sum score, while all other answers added 0 to the sum score.

#### 2.3.3. Risk Perception

According to the health belief model (HBM) [34,35], risk perception is a key effector of risk behavior [34]. It can be defined as the function of the perceived probability of an event and its expected consequences and has been assessed as the mathematical product of the perceived probability and severity of a certain outcome [35,36]. We, therefore, questioned the participants about the perceived probability of contracting a TBEV infection (E^inf^), the perceived probability of developing complications after delivery of the TBE vaccine (E^vac^), the perceived severity of natural infection from TBEV (C^inf^), and the perceived severity of vaccine side effects (C^vac^). Respondents rated the aforementioned items through a fully labeled 5-point scale (1: “almost zero”, 2: “low or rather low”, 3: “moderate”, 4: “high or rather high”, 5: “very high”), and two distinctive risk perception scores (RPS, potential range 1 to 25) for TBEV natural infection (RPS^inf^) and for TBEV vaccination (RPS^vac^) were calculated as follows:RPS^inf^ = E^inf^ × C^inf^(2)
RPS^vac^ = E^vac^ × C^vac^(3)

#### 2.3.4. Attitudes toward the TBE Vaccine

Attitudes have been defined as the learned tendency to evaluate a particular entity with some degree of favor or disfavor [37]. For the aims of this study, we specifically inquired as to the participants’ attitudes toward vaccines and immunizations, through a selected set of declarative sentences regarding the reasons to accept immunizations and, more specifically, TBE vaccination (i.e., “to avoid getting TBE”, “to avoid transmitting VPDs”, “to avoid complications of VPDs”, “to avoid VPDs in subjects who cannot be vaccinated”), or, rather, to refuse them (e.g., “to avoid shots/medications”, “uselessness”, “fear of side effects”, “religious/ethical reasons”, etc.). All items were presented as dichotomous options (yes vs. no). Even though TBEV exhibits no inter-human spreading, we deliberately included among the motivators to receive the TBE vaccine the statements “to avoid transmitting VPDs” and “to avoid VPDs in subjects who cannot be vaccinated”, in order to ascertain the degree of social desirability bias affecting the collected results.

#### 2.3.5. Preventive Practices

Participants were initially questioned regarding their TBE immunization status. Participants having reported a tick bite in the previous holiday season were questioned about their follow-up (i.e., who removed the tick head; whether any antibiotic therapy was given; whether there was any laboratory follow-up; whether the participant was then diagnosed with TBEV infection, Lyme disease, or a skin infection associated with the tick bite). The preventive measures that they had put in place were then assessed using a 5-point Likert scale, ranging from never (score = 1) to always (score = 5), and the results were dichotomized as “often” to “always” vs. “never” to “sometimes”.

### 2.4. Ethical Considerations

In order to avoid any harm to or stigma on the participants, the anonymous design of the questionnaire was guaranteed by limiting the recall of personal information and avoiding any inquiry about clinical data regarding the potential participants. As the anonymous data were analyzed for statistical and research purposes, the present study did not configure itself as a clinical trial, but rather as an opinion-based questionnaire survey. To prevent stress, anxiety, and even panic potentially caused by an inappropriate understanding of an item included in the knowledge test, correct answers and detailed explanations were then provided after completion of the questionnaire. A preventive assessment by a competent Ethical Committee and Institutional Review Board was therefore not statutorily required (Italian Official Gazette no. 76, dated 31 March 2008; European Regulation 2016/679, point no. 26) (Appendix A) [38,39].

### 2.5. Data Analysis

Continuous variables were initially reported as mean ± standard deviation (SD), and their distribution was assessed through the D’Agostino-Pearson K2 test. Variables with a K2 test *p*-value of ≥ 0.100 were considered normally distributed and were compared using Student’s *t*-test for unpaired data or an ANOVA where appropriate, while their correlation was assessed by means of Pearson’s correlation coefficient. A K2 test *p*-value of <0.100 identified those variables not normally distributed, which were analyzed using a Mann–Whitney or Kruskal–Wallis test, while their correlation was analyzed by means of Spearman’s rank test. 

Categorical variables were initially reported as percentage values. Cumulative scores (RPS^inf^, RPS^vac^, GKS, SKS) were initially normalized to a percentage value and then dichotomized by the median value as “high” vs. “low” estimates, these being managed as categorical variables. The distribution of the categorical variables was assessed through the chi-squared test with respect to the dichotomous outcome variable of having been or not been vaccinated against TBE. All variables that, in univariate analysis, were associated with an outcome variable with a *p*-value of < 0.05 were included in stepwise binary logistic regression analysis, along with the calculation of the corresponding adjusted odds ratio (aOR) and their 95% confidence intervals (95%CI).

The statistical analyses were performed using IBM SPSS Statistics 26.0 for Macintosh (IBM Corp. Armonk, NY, USA), R (version 4.3.1) [40], and Rstudio (version 2023.06.0 Build 421; Rstudio, PBC; Boston, MA, USA) software, by means of the packages epiR (version 2.0.62) and fmsb (version 0.7.5).

## 3. Results

### 3.1. Demographic Characteristics of the Participants

As shown in Figure 1, a total of 942 respondents fulfilled all the inclusion criteria (4.6% of potential recipients and 83.3% of the initial sample).

As shown in Table 1, 30.9% of participants had any occupational background in healthcare settings, while any background in agricultural settings was reported by 1.6% of participants. Around 69.1% of participants reported an educational achievement at university level, with 28.3% reporting up to 13 years of formal education. A pet within the household was reported by 43.9% of participants.

### 3.2. Knowledge Test

As shown in Table 2, a summary GKS of 58.17% ± 16.93 was calculated, with a median value of 58.33%. The corresponding SKS was estimated at 57.67% ± 17.70 (median value 57.89%). Both estimates were substantially skewed, as confirmed by the K2 test (K2 = 63.71, and 37.55, respectively; *p* < 0.001 for both summary scores) (Appendix B, Figure A2 and Figure A3). Cronbach’s alpha was estimated to be 0.824 for the knowledge test, which suggested good internal consistency (Appendix B, Table A1).

### 3.3. Risk Perception

The frequency of TBEV infection was perceived as high/very high by 51.6% of respondents, while the severity of the clinical syndrome was acknowledged as high/very high by 79.9% of respondents, with a corresponding RPS^inf^ estimate of 52.20% ± 24.28. Conversely, only 5.7% of respondents acknowledged as being somewhat high both the frequency and severity of the side effects associated with the TBE vaccine (Figure 2 and Appendix B, Table A2), with a corresponding RPS^vac^ of 11.63% ± 15.42 (Table 2 and Appendix B, Figure A2b,c). In both cases, normal distribution was eventually rejected (K2 test: *p* < 0.001).

### 3.4. Attitudes toward the TBE Vaccine

The large majority of respondents were somewhat in favor of vaccination practice (74.3%), and a total of 267 participants (28.3%) had reportedly been vaccinated against TBEV (216 out of 561 residents from a high-risk area; 38.5%). As shown in Figure 3, the peak level of vaccination coverage was reported in the age group of 60–69 years (42.6%), followed by ≥70-year-olds (38.5%), then by the age groups of 40–49 years (30.8%), 50–59 years (29.2%), 20–29 years (26.3%), and 30–39 years (17.6%). No previous TBE vaccinations were reported among those subjects under 20 years old.

A fairly large proportion of respondents (30.9%) had been recommended TBE vaccination by healthcare professionals, including personnel from the competent Local Health Unit (10.8%) and other medical professionals (i.e., their general practitioner, 6.1%; personnel from the emergency department, 1.0%; their occupational physician, 1.0%). 

Personal reasons for being or not being vaccinated against TBEV are reported in Figure 4a and Figure 4b, respectively. The most frequently reported reason for being vaccinated was identified as “avoid getting the disease” (60.7%), followed by “avoid complications” (29.2%). Conversely, “avoid spreading the disease” and “protecting those who cannot be vaccinated” were reported by 18.0% and 12.7% of participants, respectively.

When dealing with perceived barriers, 18.6% of non-vaccinated individuals (N = 675) did not know that an effective vaccine against TBE exists. Nonetheless, the most frequently reported reason for avoiding or delaying TBE vaccination was identified as not perceiving the individual risk of getting TBEV infection (24.1%), followed by not having had enough time (16.2%), the unavailability of the TBE vaccine from vaccination services (10.6%), while the fear of side effects and the lack of trust in vaccines were reported by only 3.2% and 1.3% of participants, respectively. Interestingly, only 2.4% of the non-vaccinated respondents complained about the high cost of TBE vaccination.

### 3.5. Practices

As shown in Table 3, 156 participants reported experiencing a tick bite during 2022 (16.6%), and the tick head was mostly removed by the participant himself/herself (71.2%) or by friends and relatives (21.1%). Only 7.7% of bitten participants were assisted by any HCW. The peak for tick bites occurred in individuals aged 70 years or more (38.5%), followed by the age groups of 60–69 years (21.3%), 40–49 years (20.5%), and 50–59 years (19.9%). Previous tick bites were reported by 9.9% of participants aged 30–39 years and by 6.4% of individuals aged 20–29 years, with no event occurring among individuals who were <20 years old (Appendix B, Figure A4). Antibiotic treatment was reported by 11.5% of participants, while a laboratory follow-up was performed for 19.2% of them. Interestingly, no case of TBE was identified, while 5.1% of participants reported an eventual diagnosis of Lyme disease, and a similar share of respondents was affected by any skin infection on the site of the tick bite.

Overall, 67.3% of participants reported taking at least one preventive measure against tick bites, while 25.0% reported taking up to 2 preventive measures, and 7.7% of participants reported taking 3 or more preventive measures. The single most frequently reported intervention was tucking pants into socks or boots (42.3%), followed by the use of repellent (25.0%), wearing long sleeves and pants (13.5%), wearing a hat (11.5%), avoiding typical tick habitats (11.5%), and wearing light-colored clothing (7.7%). Interestingly, checking the body was never reported. Focusing on participants not reporting any preventive measure against tick bites, their proportion decreased from 55.56% in the 30–39 age group, to 43.75% in the 40–49 age group, 27.27% in the 50–59 age group, and 10.00% in the 60–69 age group, but the proportion was 20.00% in ≥70-year-old respondents (Appendix B, Figure A5). 

### 3.6. Univariate Analysis

GKS and SKS were positively correlated (rho = 0.334, *p* < 0.001). Conversely, a negative correlation was found between RPS^vac^ and RPS^inf^ (rho = −0.078, *p* = 0.017). Moreover, both GKS and SKS were positively correlated with RPS^inf^ (rho = 0.084, *p* = 0.009, and rho = 0.262, *p* < 0.001) and were negatively correlated with RPS^vac^ (rho = −0.281, *p* < 0.001, and rho = −0.123, *p* < 0.001) (Appendix B, Table A3). In other words, a better performance in the general knowledge test was associated with a better understanding of the potential symptoms of TBE, and a better knowledge status was associated with a higher perception of the risk associated with TBEV and fewer concerns regarding the TBE vaccine.

Estimates for GKS (62.09% ± 13.19 vs. 56.38% ± 18.17; *p* < 0.001), SKS (59.63% ± 17.32 vs. 57.06% ± 17.85; *p* = 0.023), and RPS^inf^ (62.88% ± 26.41 vs. 47.95% ± 22.13; *p* < 0.001) were significantly higher among those participants who had previously been vaccinated against TBEV than among those who had not. Conversely, RPS^vac^ scores were higher among participants not reporting the previous uptake of the TBE vaccine than among vaccinated participants (6.79% ± 10.45 vs. 47.95% ± 22.13, *p* < 0.001). 

The association between the descriptive variables and TBE vaccination status is detailed in Table 4. Briefly, previous TBE vaccination was more frequently reported among individuals from high-risk areas (80.9% vs. 50.7%, *p* < 0.001), scoring better GKS (50.6% vs. 36.9% among non-vaccinated individuals, *p* < 0.001) and SKS (55.1% vs. 42.7%, *p* = 0.002), reporting a higher risk perception for natural TBEV infection (68.5% vs. 47.1%, *p* < 0.001), exhibiting a better attitude toward vaccines (in general; 96.6% vs. 83.6%, *p* < 0.001), and reporting a tick bite in the previous holiday season (33.7% vs. 9.3%, *p* < 0.001). Conversely, TBE vaccination was less frequently reported among individuals aged under 50 (50.6% vs. 61.3%, *p* < 0.001) and in those having higher RPS^vac^ scores (25.8% vs. 47.1%, *p* < 0.001).

### 3.7. Multivariable Analysis

The results of the multivariable analysis are reported in Table 5. The model included an age of <50 years, living in high-risk areas, high GKS, SKS, RPS^vac^, and RPS^inf^ scores, reporting a favorable attitude toward vaccination, and reporting any tick bite during 2022. 

In fact, being resident in high-risk areas (aOR 3.010, 95%CI 2.062 to 4.394), higher GKS (aOR 1.515, 95%CI 1.71 to 2.142) or RPS^inf^ (aOR 2.566, 95%CI 1.806 to 3.646) scores, reporting a better attitude toward vaccines (aOR 3.824, 95%CI 1.774 to 8.244), and reporting a tick bite during 2022 (aOR 5.479, 95%CI 3.582 to 8.382) were all collectively characterized as positive predictive variables. Conversely, being under 50 (aOR 0.646, 95%CI 0.458 to 0.913) and having higher RPS^vac^ scores (aOR 0.541, 95%CI 0.379 to 0.772) were characterized as negative predictive variables.

## 4. Discussion

### 4.1. Key Results

In this cross-sectional study concerning tourists’ knowledge, attitudes, and practices (KAP) regarding TBE and TBE vaccination, a total of 942 questionnaires were ultimately collected. The self-reported TBE vaccination rate was 28.3% (38.5% in people living in high-risk areas). No cases of TBE were reported, but 5.1% of respondents with a previous tick bite did allegedly receive a diagnosis of Lyme disease. TBE was acknowledged as a serious disorder by 79.9% of participants, while the likelihood of its occurrence was considered high/very high by 51.6% of them. Overall knowledge status was substantially unsatisfactory, although the inclusion criteria may reasonably have led to the oversampling of individuals with a better understanding of TBE [16,30]. Individual factors positively associated with the uptake of the TBE vaccine were identified: being resident in high-risk areas; higher GKS and RPS^inf^ scores; reporting a favorable attitude toward vaccines; and reporting a tick bite in the previous holiday season. Conversely, belonging to younger age groups and higher RPS^vac^ scores were identified as negative explanatory variables. The occurrence of tick bites (16.6% of participants) and preventive interventions allegedly being put in place were heterogeneously distributed across the age groups, with a large share of participants reporting a low proportion of protective strategies, particularly among younger individuals (Table 3 and Appendix B, Figure A4). As the younger age groups also reported low rates of tick bites, some explanations can be tentatively proposed. First, younger people often practice a different kind of tourism [41] involving lower interaction with tick-populated environments, because of the seasonality or environmental features (e.g., altitude), or sports activities different from hiking (downhill racing, mountain biking, paragliding, etc.). Moreover, age is often associated with a different understanding of the risks associated with certain behaviors. Finally, we cannot rule out a certain recall bias, which, in turn, is associated with different tourism habits. Still, as shown in Appendix B, Table A4, younger age groups were characterized by a better knowledge status and no substantial differences in terms of RPS^inf^ scores.

### 4.2. Interpretation

Although avoiding tick bites remains the most effective strategy for preventing the entirety of tick-borne diseases (e.g., Lyme disease, babesiosis, ehrlichiosis, Rocky Mountain spotted fever, anaplasmosis, or Crimean Congo hemorrhagic fever) [42,43,44], the efficacy of TBE vaccines has been well-documented [16,27,45,46]. Therefore, a rational public health approach for achieving the control of TBE in high-risk areas [16,18,27] cannot rule out interventions aimed at improving vaccination rates [24,47]. To date, Italian vaccination rates for TBE remain largely unknown [16,22,27,30]; previous KAP studies on TBE vaccines mostly focused on occupational settings [21,22], this being hardly representative of the general population because of the high background qualifications and literacy of the targeted occupational sub-group [21,48] and the well-defined geographical area (i.e., the Autonomous Province of Trento) [21,22,30] (Appendix B, Table A5).

To the best of our knowledge, here, we report on the first Italian study regarding TBE vaccine acceptance in people who may potentially be exposed to tick bites, not only within the domain of travel medicine but also among the general population living in high-risk areas [49,50,51], and our results appear to be substantially in line with previous international reports [52,53]. For instance, a recent survey from 20 European countries provided a pooled vaccination rate of 22% (range: 7% in Romania to 81% in Austria) [18,24], while a cross-sectional study from 11 European countries (year: 2018; Czech Republic, Estonia, Finland, Germany, Hungary, Latvia, Lithuania, Poland, Slovakia, Slovenia, and Sweden) identified an average vaccination rate of 25% [24,28]. Even in our previous study on farm workers from the Autonomous Province of Trento, a similar uptake of the TBE vaccine was reported (24.5%) [21].

Interestingly, the reported barriers and positive effectors regarding TBE vaccination uptake were also consistent with the available reports [24,53], which collectively stress the role of vaccine hesitancy (delay in acceptance or the refusal of vaccines despite the availability of vaccination services) [54] among the causes for low vaccination rates. While the general lack of trust in vaccines is usually considered a marginal barrier to the TBE vaccine [28], its uptake is more often affected by inappropriate TBE risk perception [23,24,28,45,46,55], particularly when dealing with its perceived incidence. Consistently with our estimates, in a previous international study, TBE was acknowledged as a severe condition by 78% of the participants, but its potential occurrence was perceived as significant by only 58% of them [28]. Similarly, a survey on travelers and travel clinics in Canada, Germany, Sweden, and the UK reported a low or even very low risk awareness of TBE, which was highly dependent on the actual prevalence of TBE in the parent country [53]. Conversely, in line with the HBM, living in an area with high TBE incidence rates is usually correlated with better vaccination propensity and vaccination rates for TBE [24,28,46,53,56], as well as having been bitten by a tick or having had any previous personal interaction with TBE cases [34,35]. However, the designation of a certain area as being at risk for TBE is sometimes unknown to the general population [24]; therefore, TBE vaccination uptake is more often modeled by the perceived endemicity of TBE than by its actual occurrence [45]. As a consequence, interventions improving general awareness and increasing media attention to TBE can considerably increase vaccine acceptance [24], particularly among individuals reporting a high frequency of visits to forests or other areas with TBE risk [53], or those who spend their leisure time in high-risk areas [28,45]. Therefore, the unsatisfactory knowledge status of participants, and its correlation with risk perception, collectively stress the potential significance of informative campaigns aimed at filling knowledge gaps in the general population [52,53]. In this regard, counseling by medical professionals has been characterized as a strong predictor of eventual vaccination [28] and should be taken into account when designing specific interventions [57,58].

A notable feature of our study was the very low proportion of respondents identifying the cost of the vaccine as a limitation, as the direct and indirect costs of TBE vaccines usually represent a potential barrier to achieving high vaccination rates [21,22,30,55]. This was somehow unexpected since in Italy, even in high-risk areas, the status of TBE vaccine pricing is quite variable. Depending on local regional recommendations, the vaccine can be delivered after either full payment or co-payment by interested citizens, or may even be totally free of charge, affecting people’s adherence to vaccination strategies. Even the recent Italian National Vaccination Plan 2023–2025 has identified some recommendations for the delivery of the TBE vaccine in occupationally exposed individuals [59,60] by including people living and traveling abroad in regions characterized by high TBE prevalence according to the WHO case definition, but the charge for this offer is still inconsistent across the various Italian regions, potentially impairing high vaccination rates in tourists visiting high-risk areas [21,25,26].

### 4.3. Limitations and Generalizability

Our study is affected by several limitations. First, the cross-sectional design cannot support conclusions on the causal relationships between the assessed risk factors and the targeted outcomes [61]; potential explanatory variables retain their significance only if they existed before the delivery of vaccine(s) [62]. As TBE vaccine uptake clearly existed before the delivery of the questionnaire, the actual role of potential explanatory variables such as knowledge status and risk perception, these being strongly influenced by personal experiences and media coverage at the time of the survey, should be quite cautiously acknowledged [63].

Second, our sample was collected by convenience sampling, and we cannot rule out some degree of self-selection bias, a substantial shortcoming shared by web-based cross-sectional studies [64] that ultimately leads to the over-sampling of individuals characterized by a particular attitude to sharing personal information through the internet and social media, a condition that is often associated with better literacy and younger age, and sharing increased interest on a specific topic [65]. Not coincidentally, our study oversampled individuals from younger age groups (60.2% of participants were younger than 50 years old at the time of the survey), the majority of participants were from geographic areas characterized by a high incidence of TBEV infections, and nearly 70% of the respondents reported having a university-level degree. Moreover, around 1/3 of the participants had an occupational background in a healthcare setting, reasonably sharing a stronger will to contribute to healthcare-related research, and probably a better understanding of this topic [22]. Studies on TBE vaccine acceptance have often reported a worse uptake of the vaccine among younger people [24,28,46]; being younger than 50 years was identified as a negative predictor of eventual acceptance of the TBE vaccine (aOR 0.646, 95%CI 0.458–0.913). This result is highly consistent with a recent study on Polish residents from high-risk areas, where the acceptance of the TBE vaccine was lesser among respondents aged 40 years or younger than among older individuals (OR 0.40, 95%CI 0.21 to 0.76) [23]. Nonetheless, since our study deliberately focused on people with previous knowledge of TBE, their better understanding of the subject has possibly inflated both the knowledge and risk perception estimates and, most notably, the reported vaccination rates. All the aforementioned potential shortcomings have been shared by other KAP surveys on healthcare topics [66,67]. Their potential role in modeling actual vaccine acceptance should, therefore, be accurately addressed [24,33,68], supporting a precautionary appraisal of our data, particularly when our estimates are compared to other reports, not only at an international [52,53] but also at a national level [68,69,70,71,72,73].

Third, the working definition of vaccination status has reasonably impaired the eventual reliability of our estimates. Compared to other immunization programs, the classical schedule of TBE vaccination is quite complicated, requiring a priming series of three doses, followed by a booster dose and several periodic shots [16,27]; the available studies on travelers suggest that up to 50% of reportedly vaccinated individuals may have failed to complete the full schedule [52,53]. As a consequence, we cannot rule out the possibility that a substantial share of the participants overstated their actual vaccination status. For example, Pilz et al. [24] have recently stressed that in sampled individuals from Poland, Hungary, and Slovakia, only 24% of persons in their study were actually protected against TBE. Nonetheless, a recent report from Pugh et al. [47] has suggested that travelers heading to high-risk areas could probably be protected against TBE for at least 5 months after two primary doses of FSME-IMMUN^®^, with the third dose still being required for achieving long-term protection. Hence, future iterations of the present study should more carefully retrieve information on the timeframe of the vaccination in order to provide more accurate estimates of actual protection against TBE.

Fourth, since the collected data were not externally validated, we cannot rule out a certain degree of declarative bias. It is plausible that some of the respondents did not truly adhere to the study requirements, having preferentially reported “socially desirable” status and attitudes instead of their actual ones. Not coincidentally, 18.0% of vaccinated responders acknowledged the aim of avoiding the spread of the disease as a key motivation for having been vaccinated against TBE, with 12.7% allegedly aiming to protect those who cannot be vaccinated. TBEV has shown no inter-human spreading [1,13], and even though this attitude may indirectly represent the solidaristic background often underlying vaccine propensity, a quite cautious appraisal of our estimates is forcibly warranted. In other words, not only did our study eventually oversample individuals with a fairly good understanding of tick-borne disorders but it is also reasonable that even the actual acceptance of vaccines and preventive interventions was similarly inflated [74].

Fifth, even though our sample largely fulfilled the minimum sample-size requirements, around 1000 respondents are relatively few when compared to the whole of the population from Italian regions as characterized by TBEV circulation, with millions of tourists each year traveling to the Dolomite Mountains; they may also be drawn from a potentially targeted population [41]. In fact, from a potential population of 20,444 individuals, our study had a 4.6% response rate, which impairs an extensive generalization of our results, particularly in a country such as Italy, which is characterized by extensive heterogeneities in health literacy. However, our sample was substantially in line, both in terms of its demographics and with the eventual results, with similar international studies, preserving its significance for comparison with the available estimates [52,53,75,76].

Sixth, due to its design, our survey does not provide any substantial insight into pediatric age groups (i.e., individuals < 18 years old). Although there is some evidence that the incidence of TBE is higher in older vs. younger children, and in adults vs. children [77], with younger age groups usually reporting a milder course of disease [77,78] and fewer neurological sequelae [77], recent reports have stressed that severe forms can also occur in children and adolescents [78], with an increasing number of cases being reported in pre-school children. As a consequence, a future follow-up of the present study should address the preventive practices implemented by parents in order to prevent TBE virus infection in their offspring, in order to better appreciate the pros and cons of pediatric vaccination strategies, at least for children living, traveling to, or with familiarity with high-risk areas [16,27,77,78]. 

Last but not least, previous studies have stressed the importance of media coverage of TBEV infections [45]. By raising the eventual risk perception about an otherwise often misunderstood infection, the eventual impact of new and social media on the acceptance of the TBE vaccine may even exceed that of other topics of public health interest [79]. As a proxy of the media coverage on TBE and other tick-borne infections, we specifically explored the relative search volumes provided by Google Trends™ [80,81]. As shown in Appendix B, Table A6, during the study period, no correlation was found for RPS^inf^, RPS^vac^, and their components. In other words, risk perception on TBE, TBEV infections, and tick-borne diseases among the study participants was seemingly not influenced by new media coverage.

## 5. Conclusions

Our study suggests that tourists to the Dolomite Mountains, a high-risk area for TBE and TBEV virus infection, collectively exhibit an extensive lack of knowledge of this pathogen, with inappropriate risk awareness. Moreover, actual uptake of the TBE vaccine, while still in line with European data, was largely unsatisfactory. As TBEV infection may be effectively countered through a combination of effective behavioral practices and TBE vaccination for people either living or performing outdoor activities in high-risk areas, 69 improving overall awareness of these preventive options could, therefore, be instrumental in reducing the eventual burden of TBEV infections (see Appendix B, Table A7). 

## Figures and Tables

**Figure 1 tropicalmed-08-00491-f001:**
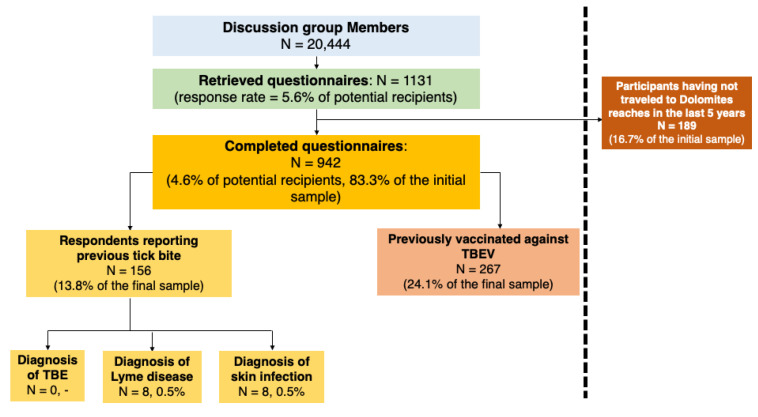
Flow chart for the participants included in the final sample.

**Figure 2 tropicalmed-08-00491-f002:**
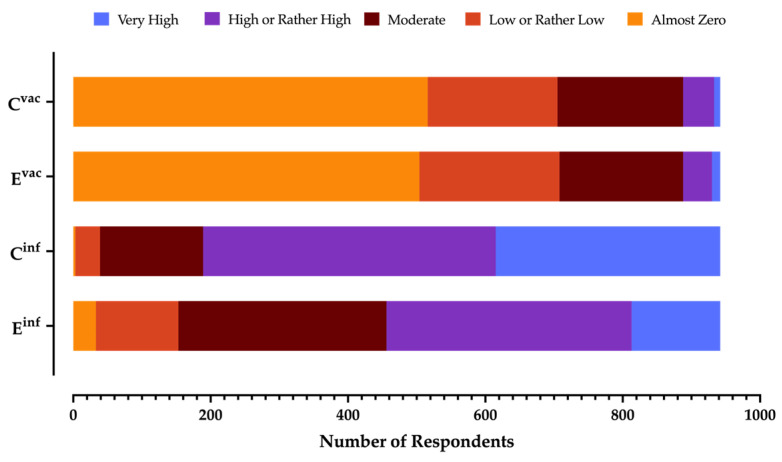
Summary of the perceived probability of contracting a TBEV infection (E^inf^); perceived probability of developing complications after the delivery of the TBE vaccine (E^vac^); perceived severity of natural infection from TBEV (C^inf^); perceived severity of the vaccine side effects (C^vac^), as reported by 942 participants (Italy, 2023).

**Figure 3 tropicalmed-08-00491-f003:**
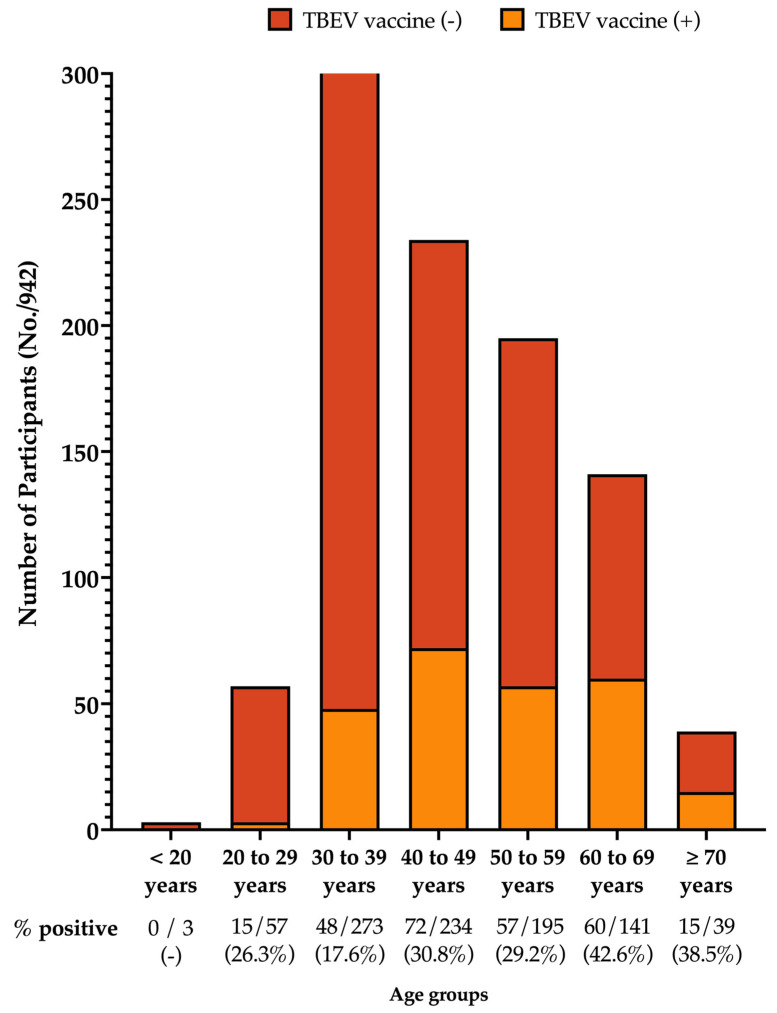
Vaccination rates for TBE according to age group, as reported by 942 participants (Italy, 2023).

**Figure 4 tropicalmed-08-00491-f004:**
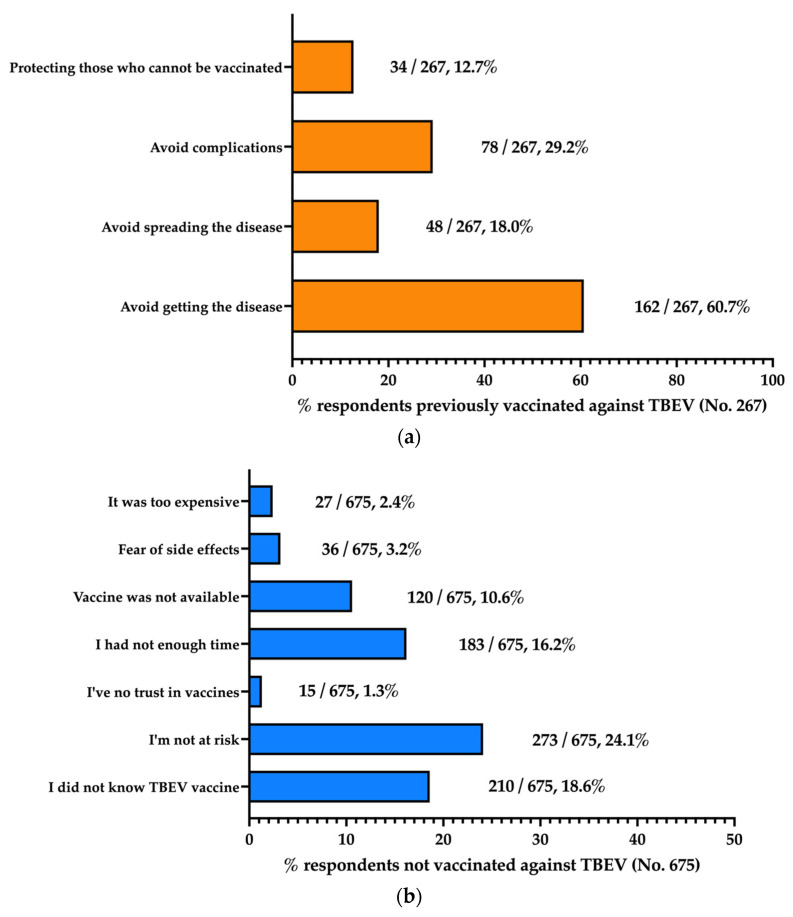
Reasons for receiving (**a**) and not receiving (**b**) the TBEV vaccination protocol, as reported by 942 participants (Italy, 2023).

**Table 1 tropicalmed-08-00491-t001:** Characteristics of 942 tourists to the Dolomites area who participated in this survey (Italy, 2023).

Variables	N/942, %
Age (years)	
<20	3, 0.3%
20–29	57, 6.1%
30–39	273, 29.0%
40–49	234, 24.8%
50–59	195, 20.7%
60–69	141, 15.0%
≥70	39, 4.1%
Male gender	369, 39.2%
Region of Origin	
High-risk Areas for TBEV ^1^	561, 59.6%
Northern Italy ^2^	351, 37.3%
Central Italy ^3^	15, 1.6%
Southern Italy ^4^	12, 1.3%
Major Island ^5^	3, 0.3%
Any occupational background in healthcare settings	291, 30.9%
Any occupational background in agricultural settings	15, 1.6%
Educational achievement	
University (≥14 years of formal education)	651, 69.1%
High school (9–13 years of formal education)	267, 28.3%
Primary/Secondary school (≤8 years of formal education)	24, 2.5%
Any pet in the household	414, 43.9%
GKS > median value (58.33%)	396, 42.0%%
Symptom score > median value (57.89%)	441, 46.8%
RPS for tick-borne infections > median value (49.00%)	465, 49.4%
RPS for TBE vaccine > median value (6.00%)	405, 43.0%
Tick bite in the previous season	156, 16.6%
Attitudes Toward Vaccines	
In general, somehow favorable	840, 74.3%
Previously vaccinated against TBEV	267, 28.3%
TBEV was recommended by any healthcare professional	291, 30.9%
General Practitioner	57, 6.1%
Medical Officer from Vaccination Services of the Local Health Unit	102, 10.8%
Emergency Department	9, 1.0%
Occupational physician	9, 1.0%
Other non-medical HCWs	114, 12.1%

Note: TBEV = tick-borne encephalitis virus; HCWs = healthcare workers; GKS = general knowledge score; RPS = risk perception score; ^1^ = Autonomous Province of Trento, Province of Belluno, and Province of Gorizia; ^2^ = Piedmont, Liguria, Lombardy, Emilia-Romagna, Autonomous Province of Bolzano, Veneto (excluding the province of Belluno), and Friuli-Venezia-Giulia (excluding the province of Gorizia); ^3^ = Tuscany, Umbria, Marche, and Latium; ^4^ = Campania, Abruzzo, Basilicata, Molise, Apulia, and Calabria; ^5^ = Sicily and Sardinia.

**Table 2 tropicalmed-08-00491-t002:** Summary of general knowledge score, symptom knowledge score, and risk perception score per tick-borne infection and for the tick-borne encephalitis virus (TBEV) vaccine.

Cumulative Score	Average ± SD	Median (Range; Min–Max)	K2 (*p*-Value)
General Knowledge Score	58.17 ± 16.93	58.33 (1.00; 100)	K2 = 63.71, *p* < 0.001
Symptom Knowledge Score	57.67 ± 17.70	57.89 (15.79; 100)	K2 = 37.55, *p* < 0.001
Risk Perception Score, Tick-borne Infections	52.20 ± 24.28	49.00 (1.00; 100)	K2 = 30.11, *p* < 0.001
Risk Perception Score, TBE Vaccine	11.63 ± 15.42	6.00 (1.00; 100)	K2 = 569.1, *p* < 0.001

**Table 3 tropicalmed-08-00491-t003:** Characteristics of the tick bites, as reported by 156 participants.

Variable	N/156, %
Tick Removed by:	
Any HCW	12, 7.7%
Friends/Relatives	33, 21.1%
Him-/Herself	111, 71.2%
Antibiotic treatment	18, 11.5%
Laboratory follow-up	30, 19.2%
Diagnosis of Lyme disease	8, 5.1%
Diagnosis of TBEV infection	0, -
Diagnosis of skin infection on tick bite	8, 5.1%
Preventive Measures (Often/Always)	
Use of Repellent	39, 25.0%
Wear light-colored clothing	12, 7.7%
Wear long sleeves and pants	21, 13.5%
Tuck pants into socks or boots	66, 42.3%
Perform body check	0, -
Wear a hat	18, 11.5%
Avoid typical tick habitats	18, 11.5%
Any	105, 67.3%
1 preventive measure	54, 34.6%
2 preventive measures	39, 25.0%
3 preventive measures or more	12, 7.7%

Notes: HCW = healthcare worker; TBE = tick-borne encephalitis; TBEV = TBE virus.

**Table 4 tropicalmed-08-00491-t004:** Univariate analysis of the individual factors associated with previous vaccination against tick-borne encephalitis virus (TBEV). Comparisons were performed by means of the chi-squared test for categorical values and the Mann–Whitney U test for continuous variables.

All Respondents	Previously Vaccinated Against TBEV		
	Yes (N/267, %)	No (N/675, %)	*p*-Value
General Knowledge Score (%, average ± SD)	62.09 ± 13.19	56.38 ± 18.17	<0.001
Symptoms Score (%, average ± SD)	59.63 ± 17.32	57.06 ± 17.85	0.023
Risk Perception Score, Tick-Borne infection (%, average ± SD)	62.88 ± 26.41	47.95 ± 22.13	<0.001
Risk Perception Score, TBE vaccine (%, average ± SD)	6.79 ± 10.45	47.95 ± 22.13	<0.001
Age < 50 years	135, 50.6%	414, 61.3%	<0.001
Male gender	105, 39.3%	255, 37.8%	0.925
Resident in high-risk areas	216, 80.9%	342, 50.7%	<0.001
Any occupational background in healthcare settings	93, 34.8%	198, 29.3%	0.177
Any occupational background in agricultural settings	6, 2.2%	6, 0.9%	0.106
High educational achievement	174, 65.2%	459, 68.0%	0.136
Any pet in the household	126, 47.2%	285, 42.2%	0.345
Higher general knowledge score	135, 50.6%	249, 36.9%	<0.001
Higher knowledge of symptoms score	147, 55.1%	288, 42.7%	0.002
High risk perception regarding tick-borne infections	183, 68.5%	273, 40.4%	<0.001
High risk perception regarding TBE vaccine	69, 25.8%	318, 47.1%	<0.001
Attitude toward vaccines (favorable)	258, 96.6%	564, 83.6%	<0.001
Tick bite in the previous season	90, 33.7%	63, 9.3%	<0.001

Note: high-risk areas = Autonomous Province of Trento, the Province of Belluno, and the Province of Gorizia; SD = standard deviation.

**Table 5 tropicalmed-08-00491-t005:** Multivariable analysis of the factors associated with previous vaccination against tick-borne encephalitis virus (TBEV). Multi-adjusted odds ratios (aOR) and their corresponding 95% confidence intervals (95%CI) were calculated by means of two distinct binary logistic regression models, which included all factors that were associated in the univariate analysis with an outcome variable (*p* < 0.05).

All Respondents	Previously Vaccinated Against TBEV	
	aOR	95% Confidence Interval
Age < 50 years	0.646	0.458; 0.913
Living in high-risk areas	3.010	2.062; 4.394
Higher general knowledge score	1.515	1.071; 2.142
Higher knowledge of symptoms score	1.009	0.714; 1.427
High risk perception regarding tick-borne infections	2.566	1.806; 3.646
High risk perception regarding TBE vaccine	0.541	0.379; 0.772
Attitude toward vaccines (favorable)	3.824	1.774; 8.244
Tick bite in the previous season	5.479	3.582; 8.382

Note: high-risk areas = Autonomous Province of Trento, the Province of Belluno, and the Province of Gorizia.

## Data Availability

Data are available on request.

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
