# Peer review of "Tick-Borne Encephalitis Virus Vaccination among Tourists in a High-Prevalence Area (Italy, 2023): A Cross-Sectional Study"

_tropicalmed, 2023, doi:10.3390/tropicalmed8110491_

Round 1

Reviewer 1 Report (Previous Reviewer 3)

Comments and Suggestions for Authors

All my comments have been addressed, and the paper is now good for publication.

Author Response

Estimated Rev.1,

we warmly thank you for your positive appraisal.

Thank you again.

MR

Reviewer 2 Report (Previous Reviewer 2)

Comments and Suggestions for Authors

I’d like to thank the authors for their revisions. I have some further concerns regarding the revised paper. Unfortunately, making prior knowledge of TBE necessary for study inclusion is a substantial flaw which limits the applicability of the study; it almost certainly increases the apparent vaccination coverage – likely very substantially. It also very likely increases the relative knowledge scores.

However, the study still provides some data for a country where relatively few data regarding vaccination coverage and attitudes toward vaccination are available.

Important though, is that the total text is very long and should be cut down to improve readability. There are also over 100 references, which is better suited to a review article and far too many for such a study (many journals only permit approximately 35). Furthermore, it is necessary to have the text proof-read to improve the grammar – it is not yet suitable for publication in its current form.

Comments on the Quality of English Language

Please see the previous section; editing is required.

Author Response

Estimated Rev. 2,

we thank you for your comments and recommendations. Eventually, we agree that the text has excessively swollen because of the various improvements promoted by the revision process. Even though MDPI's instructions to the Authors does not include any word or reference limit, we've amended the present text by reducing its overall length: at the moment, the total word count has decreased to 5700 for the main text (introduction to discussion), while the total number of references has been decreased to < 90. 

As you can see, the main text has been extensively revised, with the aim of improving its overall quality and readability.

In summary, we again thank you for the valuable recommendations, and hope that this revised version could be eventually accepted for publication.

Reviewer 3 Report (Previous Reviewer 1)

Comments and Suggestions for Authors

The authors have thoroughly revised the paper as suggested. No additional comments.

Author Response

Estimated Rev.3,

we thank you for the positive appraisal of our study.

MR

This manuscript is a resubmission of an earlier submission. The following is a list of the peer review reports and author responses from that submission.

Round 1

Reviewer 1 Report

Comments and Suggestions for Authors

This KAP study is of some interest. Overall, it is very long, for instance the introduction includes much information which all (should) know and even some information which is irrelevant: Why mention that there are 6 TBE vaccines as in Western Europe we only have two? On the other hand paediatric TBE is neglected as questionnaires were only offered to volunteers ≥18y (why?). The English is sometimes clumsy and needs to be revised by a native speaker — who hopefully also will streamline the text, eliminating unnecessary words, such as 'moreover', 'interestingly', etc..

37-40 Clearly misleading: TBE is not caused by 'several viral pathogens' but by a single one which has different subtypes (as you mention later). To my memory in Italy there is only TBE-Eu. Thus, delete YF, WNV, JE, etc., which are mosquito-borne and totally irrelevant for this manuscript.

46 Suggest you mention that usually there are two phases, first the non-specific, usually an interval without symptoms, and then the CNS disease in a minority of patients.

54 share > better proportion (also 311)

63 Specify please: is that for Italy or for specific province(s)? Only the latter is relevant

104 Add please: what is the maximum altitude of the Dolomites and compare to the maximum altitude where you can find ticks there

124/131/251 To make it clearly understandable suggest in the green bar of Fig. 1 to specify in the second line: (response rate 5.6% = potential recipients)

Fig. 1 It is very unusual in English to speak of 'No.' > N = is better

259 Delete (males = 39.2%) as that can be seen in Table 1, avoid repetition

Table 1 Why 'Major Island'? Do you also have 'Minor Islands'? Is that relevant? Why not just state 'Island'?

408 You refer to the 'general population' and also to exposure and vaccine coverage in professional groups in your discussion, while actually the target population is visitors, tourists in the Dolomites. Thus overall you should rather make (or at least add) a link to TBE travel studies.

455 Typo: primer > primary schedule as compared to booster doses

511 Limits > Limitations

512ff The low response rate must explicitly be mentioned. Additionally would also note that paediatric TBE was neglected in the study, but that this is clinically relevant in view of subsequent cognitive disorders.

560 As per recent publications, those who have received 2 vaccine doses have a vaccine efficacy of ±90% which is high as compared to many other vaccines — thus that sentence is to be revised

592 What do you mean by 'TBE and TBEV virus infection'? Anyway, TBEV = TBE virus. Suggest 'TBE and other tick-borne infections' (to give a hint at borreliosis, which of course is not viral).

594 We usually refer to TBE vaccines, not TBEV vaccines

598 Cost-effective? Add reference.

610 Presume you can add 'written'

Comments on the Quality of English Language

See above, made a few suggestions for amendments — but it is not my task to perform close reading with correcting throughout.

Author Response

Estimated Reviewer,

we would thank you for the accurate and collaborative review you’ve provided, whose content has likely contributed to the significant improvement of our article. In the following lines we provide a point-to-point reply to your comments. Again, thank you for your contribution.

MR

Rev. 1

Estimated Reviewer,

we would thank you for the accurate and collaborative review you’ve provided, whose content has likely contributed to the significant improvement of our article. In the following lines we provide a point-to-point reply to your comments. Again, thank you for your contribution.

MR

Overall, it is very long, for instance the introduction includes much information which all (should) know and even some information which is irrelevant: Why mention that there are 6 TBE vaccines as in Western Europe we only have two? On the other hand paediatric TBE is neglected as questionnaires were only offered to volunteers ≥18y (why?). The English is sometimes clumsy and needs to be revised by a native speaker — who hopefully also will streamline the text, eliminating unnecessary words, such as 'moreover', 'interestingly', etc..

Thank you for your suggestions. In fact, we’ve simplified the introduction (and other sections as well). For instance: mention to the 6 TBE vaccines has been removed. The English text has been extensively revised and we hope the its clumsiness (we totally agree with your initial evaluation) has been at least sufficiently removed.

Regarding the pediatric TBE we’ve added a specific section in discussion:

A further limit of our study is that, by its design, our survey does not provide any substantial insight on the pediatric age groups (i.e. individuals < 18 years-old). Although there is some evidence that the incidence of TBE is higher in older vs. younger children, and in adults vs. children [104], with younger age groups usually reporting a milder course [104,105], and fewer neurological sequelae [104], recent reports have stressed that severe forms can also occur in children and adolescents [105], with an increasing number of cases reported in pre-school children. As a consequence, a future follow-up of the present study should address preventive practices implemented by parents in order to prevent TBE virus infection in their offspring, in order to better appreciate pros and cons of pediatric vaccination strategies, at least for children living, travelling to, or with fa-miliarity with high-risk areas [14,20,104,105].

37-40 Clearly misleading: TBE is not caused by 'several viral pathogens' but by a single one which has different subtypes (as you mention later). To my memory in Italy there is only TBE-Eu. Thus, delete YF, WNV, JE, etc., which are mosquito-borne and totally irrelevant for this manuscript.

Thank you for your note. We followed the recent review of Jonhnson et al. doi:  10.1097/QCO.0000000000000924

“ Tick-borne encephalitis is an acute infection caused by a number of viruses, many within the family Flaviviridae and genus Flavivirus”. However, we totally agree with your comment, and we’ve amended the main text as follows:

Tick-Borne Encephalitis (TBE) is an acute clinical syndrome caused by a RNA virus belonging to the family of Flaviviridae, genus Flavivirus [1–3], that targets the central nervous system (CNS) and can result in long-term neurological symptoms, and even death [3–6]. The cause of TBE is the TBE virus (TBEV) [1,7–9], which includes at least five subtypes (European, TBEV-Eu; Siberian, TBEV-Sib; TBEV Far Eastern, TBEV-FE; and the recently characterized Baikalian [TBEV-Bkl] and Himalayan [TBEV-Him] subtypes). Nearly all incident cases in Europe are associated with TBEV-Eu subtype, transmitted by Ixodes  icinus ticks [1,3]. Around two thirds of TBEV infections are asymptomatic [10,11]; while symptomatic infections mostly develop a two-phased illness. The first phase is associated with non-specific symptoms including fever and malaise. The second phase may occur in a reduced number of patients, having the pathogen enter into the CNS and causing complications such as meningitis, meningoencephalitis and myelitis [1,12,13]

46 Suggest you mention that usually there are two phases, first the non-specific, usually an interval without symptoms, and then the CNS disease in a minority of patients.

Thank you for your suggestion. We’ve amended the main text as follows:

… symptomatic infections mostly develop a two-phased illness. The first phase is associated with non-specific symptoms including fever and malaise. The second phase may occur in a reduced number of patients, having the pathogen enter into the CNS and causing complications such as meningitis, meningoencephalitis and myelitis [1,12,13]…

54 share > better proportion (also 311)

Thank you, fixed (please note, as for following text correction that we’ve extensively amended the main text, so both the row numbers and the very same main text could have been changed)

63 Specify please: is that for Italy or for specific province(s)? Only the latter is relevant

Thank you. We’ve removed a section that in fact did not provide any significant information and focused on the regions interested by our intervention.

104 Add please: what is the maximum altitude of the Dolomites and compare to the maximum altitude where you can find ticks there

Thank you for your suggestion. We ave amended the section 2.1 as follows:

The Dolomites (Italian: Dolomiti) are a mountain range in the North-Eastern Italy, between the Austrian border in the North, and the Venetian plain in the South, whose highest peak (the Marmolada mountain) rises to around 3,343 m above the sea level. Their area (15,942 km2) is shared by the Italian Regions of Veneto, Trentino-Alto Adige/Südtirol and Friuli Venezia Giulia (provinces of Belluno, Vicenza, Verona, Trento, Bolzano, Udine, and Pordenone). The climate of the Dolomites is typically alpine, characterized by harsh winter temperatures, and heavy rainfall concentrated in the summer period. Compared to the border regions of Austria and Switzerland, Dolomites are characterized by warmer temperatures and less precipitation, being also highly accessible both in cold and warm season. These features, as well as being one of the 218 natural UNESCO world heritage sites, collectively contribute to their wide tourism attractivity, with way more than 20 million overnight stays per year. To date, ticks are usually found at altitudes < 1,500 m above sea level, from spring to autumn, at temperatures above 10°C, even though global warming is progressively raising this altitude level [26,29,30].

124/131/251 To make it clearly understandable suggest in the green bar of Fig. 1 to specify in the second line: (response rate 5.6% = potential recipients)

Thank you, the figure was amended accordingly.

Fig. 1 It is very unusual in English to speak of 'No.' > N = is better

Thank you, the figures and the tables have been amended accordingly.

259 Delete (males = 39.2%) as that can be seen in Table 1, avoid repetition

Thank you, done.

Table 1 Why 'Major Island'? Do you also have 'Minor Islands'? Is that relevant? Why not just state 'Island'?

Thank you for your suggestion. In fact, in Italy Sicily and Sardinia (both large islands from the Mediterranean Sea) have large populations, exceeding 1 million inhabitants. Therefore, Sicily and Sardinia are usually acknowledged as “Major Italian Islands” compared to “minor islands” (e.g. Isola d’Elba), and are usually acknowledged as “Minor Islands” or “Small Islands” also at administrative levels. See for example the following paper:  https://pubmed.ncbi.nlm.nih.gov/35735753/

408 You refer to the 'general population' and also to exposure and vaccine coverage in professional groups in your discussion, while actually the target population is visitors, tourists in the Dolomites. Thus overall you should rather make (or at least add) a link to TBE travel studies.

Thank you. We clearly missed the point across the various revisions of the paper. We have specifically included and discussed the references to two major papers on this topic, namely: Marano et al. (2018) 10.1093/jtm/tay063; Poulos et al. (2022) 10.1080/14760584.2022.2108798.

The main text was amended accordingly. For example:

Even though there is a certain lack of evidence about KAP on TBEV and TBE vaccines on tourists and travelers, our results are substantially in line with previous international reports[49,50]. For instance, a survey on travelers and travel clinics from Canada, Germany, Sweden and UK reported a low or even very low risk awareness on TBE, highly dependent on the actual prevalence of TBEV infections [50].

455 Typo: primer > primary schedule as compared to booster doses

Thank you, we fixed accordingly.

511 Limits > Limitations

Fixed, thank you.

512ff The low response rate must explicitly be mentioned. Additionally would also note that paediatric TBE was neglected in the study, but that this is clinically relevant in view of subsequent cognitive disorders.

Thank you again. We’ve included the following section:

Fifth, even though our sample largely fulfilled the minimum sample size re-quirements, around 1,000 respondents are relatively few when compared to the whole of the population from Italian regions characterized by TBEV circulation, and with the millions of tourists that each year travelling to the Dolomites mountains, but also with the potentially targeted population. In fact, from a potential population of 20,444 travelers, our study eventually encompassed only 942 individuals, with a 4.6% response rate, which impairs an extensive generalization of our results. However, it should be stressed that our sample was substantially in line, both in its demographics and with the eventual results, with similar international studies, preserving its significance for the comparison with available estimates [49,50,101–103].

560 As per recent publications, those who have received 2 vaccine doses have a vaccine efficacy of ±90% which is high as compared to many other vaccines — thus that sentence is to be revised

Thank you. We’ve added the reference:

Nonetheless, a recent report from Pugh et al. [98] has suggested that travelers heading to high-risk area could be likely protected against TBE for at least 5 months after two primary doses of FSME-IMMUN® (the only one formulate that has been licensed in Italy, to date), with the third dose still requested for achieving long-term protection. Hence, future it-erations of the present study should more carefully retrieve the timeframe of the vac-cination in order to provide more accurate estimates of actual protection against TBE.

592 What do you mean by 'TBE and TBEV virus infection'? Anyway, TBEV = TBE virus. Suggest 'TBE and other tick-borne infections' (to give a hint at borreliosis, which of course is not viral).

Thank you, we’ve solved the inconsistence and corrected the main text as you did suggest.

594 We usually refer to TBE vaccines, not TBEV vaccines

Thank you, we’ve fixed accordingly.

598 Cost-effective? Add reference.

We’ve modified the sentence as, in effect, the theme of cost-effectiveness had to be preventively discussed across the main text, being in fact outside the aims and the topic of this study.

610 Presume you can add 'written'

Yes, thank you again.

Reviewer 2 Report

Comments and Suggestions for Authors

In the paper “Tick Borne Encephalitis Virus vaccine among tourists in high-prevalence area (Italy, 2023): a cross-sectional study” by Ricco, et al, the authors utilized a web-based questionnaire to assess TBE vaccine uptake and tick-related knowledge, attitudes, and protective behaviors among tourists visiting the high TBE risk area in the Dolomites mountains, Italy. They found that living in higher risk areas, more knowledge of TBE, greater risk perception of tickborne diseases and more positive attitudes towards vaccines were associated with increased TBE vaccine uptake while being younger than 50 and having a greater risk perception of the TBE vaccine were associated with lower TBE vaccine uptake. These findings are in line with existing research on TBE vaccination coverage and factors influencing uptake throughout Europe. An important contribution of this paper to the field is the data from Italy, where fewer TBE vaccination coverage data are available, despite the increasing risk of TBE in the country over time. The study’s findings are also important for travelers/tourists. In principle, I think this is a nicely done paper (given the limitations inherent in web-based surveys, which the authors do nicely address). I have only a few comments to address:

The first sentence of the introduction “Tick-Borne Encephalitis (TBE) is an acute clinical syndrome caused by several viral pathogens, many of them belonging to the family of Flaviviridae…” is somewhat misleading. While there are certainly other tickborne encephalitic illness, the term Tickborne Encephalitis (TBE) itself refers only to disease caused by the Tickborne Encephalitis Virus (TBEV). Please consider rephrasing so as to avoid confusion.

In the “Study Design and Sample Size” section, the authors mention the “annual incidence of tick bite equals to 30.2%”, but use 0.32 in their sample size calculation that follows – is one of these values a typo?

From the description it seems like having previous knowledge of TBE was one of the study’s inclusion criteria? It doesn’t make sense to exclude individuals that don’t know what TBE is (accordingly, these individuals are also less likely to be vaccinated, which would bias the calculated vaccination coverage). Could the authors please clarify what is intended here?

I thought that Figure 4a (Reasons for having received TBE vaccination) was a bit strange – as the disease is not transmitted between individuals what did the authors intend to demonstrate by including “Protecting those who cannot be vaccinated” and “Avoid spreading the disease” as potential answers?

While not a science-related comment, as an important target audience for this paper could be physicians, I don’t think it could hurt to consider emphasizing already in the abstract and introduction that the Dolomites mountains are a popular tourist destination (I liked the part in the methods mentioning that over 20 million overnight stays occur there per year!) and the relevance for travel medicine.

Comments on the Quality of English Language

There are several grammatical errors, please consider having a native speaker review the paper or the use of an editorial service.

Author Response

Rev. 2

Estimated Reviewer,

we would thank you for the accurate and collaborative review you’ve provided, whose content has likely contributed to the significant improvement of our article. In the following lines we provide a point-to-point reply to your comments. Again, thank you for your contribution.

MR

The first sentence of the introduction “Tick-Borne Encephalitis (TBE) is an acute clinical syndrome caused by several viral pathogens, many of them belonging to the family of Flaviviridae…” is somewhat misleading. While there are certainly other tickborne encephalitic illness, the term Tickborne Encephalitis (TBE) itself refers only to disease caused by the Tickborne Encephalitis Virus (TBEV). Please consider rephrasing so as to avoid confusion.

Thank you for your note. We followed the recent review of Jonhnson et al. doi:  10.1097/QCO.0000000000000924

“ Tick-borne encephalitis is an acute infection caused by a number of viruses, many within the family Flaviviridae and genus Flavivirus”. However, we totally agree with your comment, and we’ve amended the main text as follows:

Tick-Borne Encephalitis (TBE) is an acute clinical syndrome caused by a RNA virus belonging to the family of Flaviviridae, genus Flavivirus [1–3], that targets the central nervous system (CNS) and can result in long-term neurological symptoms, and even death [3–6]. The cause of TBE is the TBE virus (TBEV) [1,7–9], which includes at least five subtypes (European, TBEV-Eu; Siberian, TBEV-Sib; TBEV Far Eastern, TBEV-FE; and the recently characterized Baikalian [TBEV-Bkl] and Himalayan [TBEV-Him] subtypes). Nearly all incident cases in Europe are associated with TBEV-Eu subtype, transmitted by Ixodes  icinus ticks [1,3]. Around two thirds of TBEV infections are asymptomatic [10,11]; while symptomatic infections mostly develop a two-phased illness. The first phase is associated with non-specific symptoms including fever and malaise. The second phase may occur in a reduced number of patients, having the pathogen enter into the CNS and causing complications such as meningitis, meningoencephalitis and myelitis [1,12,13]

In the “Study Design and Sample Size” section, the authors mention the “annual incidence of tick bite equals to 30.2%”, but use 0.32 in their sample size calculation that follows – is one of these values a typo?

Thank you! In fact, we did an annoying mistake across the various versions of our study and 30.2% (i.e. 0.302) became “0.32” as a consequence of a transcription error. We fixed the text and revised all estimates accordingly.

From the description it seems like having previous knowledge of TBE was one of the study’s inclusion criteria? It doesn’t make sense to exclude individuals that don’t know what TBE is (accordingly, these individuals are also less likely to be vaccinated, which would bias the calculated vaccination coverage). Could the authors please clarify what is intended here?

Thank you for your note. In fact, being our study focused on barriers and motivators for TBE vaccine, in a stark difference with other studies (e.g. Marano et al. (2018) 10.1093/jtm/tay063; Poulos et al. (2022) 10.1080/14760584.2022.2108798) where preventive measures against tick bites were initially addressed, with subsequent analysis of TBE vaccine (see for example Figure 1 of Marano et al., and Figure 2 of Poulos et al.). A more appropriate approach (as for the aforementioned study) would have required to inquiry also the factors leading to a better knowledge of TBE. We therefore opted for the present approach in order to avoid the potential collinearity issue (i.e. knowing that TBE does exists reasonably leads to being vaccinated). The potential bias has been more extensively addressed in the discussion.

For instance, our study deliberately focused on people having a previous knowledge of TBE, and their better understanding of the subject has possibly inflated both knowledge and risk perception estimates, and most notably the reported vaccination rates. As a consequence, when our estimates are compared to similar international reports, a precautionary approach is warranted [49,50].

I thought that Figure 4a (Reasons for having received TBE vaccination) was a bit strange – as the disease is not transmitted between individuals what did the authors intend to demonstrate by including “Protecting those who cannot be vaccinated” and “Avoid spreading the disease” as potential answers?

Thank you for your note. In fact, we missed the following section of the main text:

In this regard, as well as for the items included in the knowledge tests, some of the participants may have reported “socially desirable” attitudes instead of their actual ones. For instance, 18.0% of vaccinated responders acknowledged the aim of avoiding the spread of the disease as a key motivation for having been vaccinated against TBE. TBEV has no inter-human spreading, and even though this statement may indirectly represent the solidaristic background often with vaccine propensity, a quite cautious appraisal of our estimates is forcibly warranted. In other words, not only did our study eventually oversample individuals with a reasonably inappropriate understanding of tick-borne disorders, but it is reasonable that the eventual sample inflated the actual acceptance of vaccines and preventive interventions [99,100].

While not a science-related comment, as an important target audience for this paper could be physicians, I don’t think it could hurt to consider emphasizing already in the abstract and introduction that the Dolomites mountains are a popular tourist destination (I liked the part in the methods mentioning that over 20 million overnight stays occur there per year!) and the relevance for travel medicine.

Thank you! We amended the abstract (and the main text as well) as follows:

Abstract:

Therefore, we conducted a questionnaire-based survey to assess knowledge, attitudes, and preventive practices among in a convenience sample Italian tourists to the Dolomites mountains, a UNESCO world heritage site and tourist attraction with over 20 million overnight stays per year.

Main text:

The Dolomites (Italian: Dolomiti) are a mountain range in the North-Eastern Italy, between the Austrian border in the North, and the Venetian plain in the South, whose highest peak (the Marmolada mountain) rises to around 3,343 m above the sea level. Their area (15,942 km2) is shared by the Italian Regions of Veneto, Trentino-Alto Adige/Südtirol and Friuli Venezia Giulia (provinces of Belluno, Vicenza, Verona, Trento, Bolzano, Udine, and Pordenone). The climate of the Dolomites is typically alpine, characterized by harsh winter temperatures, and heavy rainfall concentrated in the summer period. Compared to the border regions of Austria and Switzerland, Dolomites are characterized by warmer temperatures and less precipitation, being also highly accessible both in cold and warm season. These features, as well as being one of the 218 natural UNESCO world heritage sites, collectively contribute to their wide tourism attractivity, with way more than 20 million overnight stays per year. To date, ticks are usually found at altitudes < 1,500 m above sea level, from spring to autumn, at temperatures above 10°C, even though global warming is progressively raising this altitude level [26,29,30].

Reviewer 3 Report

Comments and Suggestions for Authors

See attached Word document.

Comments on the Quality of English Language

See my specific comments in the Word document.

Author Response

Rev. 3

Estimated Reviewer,

we would thank you for the accurate and collaborative review you’ve provided, whose content has likely contributed to the significant improvement of our article. In the following lines we provide a point-to-point reply to your comments. Please note that, having been the main text extensively reworked, some of the corrections have been either removed or moved in other sections (for example: the text that was in row 86 now does not contain any reference to the 6 TBEV vaccines: “To date, two kinds of TBE vaccine are commercially in Western Europe, i.e. FSME-IMMUN® (Pfizer) and ENCEPUR® (Bavarian Nordic), but only the former has been licensed in Italy [14]. Both immunizations are considered effective (effectiveness ranging between 96% to 99%), and can be used in adults and children ≥ 1 year-old [14,28].”). Therefore, in the following point-to-point reply we’ve focused on the conceptual one or on those where the text was extensively reworkerd.

Again, thank you for your contribution.

MR

Table 1: keep the table title concise and move the texts in the bracket to the ‘Note’ under the table

Thank you, we did it accordingly.

Table 2move the Note to the bottom of the table and complete the title “Summary Scores of ...”

Thank you, we did it accordingly.

Line 337 – 341: older population had a higher proportion of ‘reporting’ tick bites than younger people. However, interestingly, more of them reported taking preventive measures compared to younger people. So, you concluded that ‘younger people were less frequently bitten by ticks although not regularly implementing protective strategies. (Line 426 427)’ I don’t think this conclusion is proper. Is it possible that younger people didn’t report tick bite events because they lacked knowledge to recognize or didn’t realize they were bitten by ticks?

Thank you for your note. In fact, we have included the following section:

Some explanations may be tentatively proposed. First, younger people may practice a different kind of tourisms, with lower interaction with tick-populated environments because of the seasonality or environmental features (e.g. altitude), or sport activities different from hiking (downhill, mountain biki, paragliding, etc.). Moreover, it is plau-sible that younger age groups may have a different understanding of the risks associated with tick bites, with resulting recall bias. However, as shown in Annex Table A7, younger age groups were characterized by a better knowledge status, and no substantial dif-ferences when dealing with RPSinf.

Also the newly designed Annex Table A7 was included.

Table 3: move the Notes to the bottom of the table.

Thank you, we did it accordingly.

Table 4: move the methodological descriptions to the bottom of the table as ‘Notes’ to keep the table title concise.

Thank you, we did it accordingly.

Line 393: since you already calculated the adjusted odds ratio in your ‘Univariate analysis’ part, I don’t see the significance of doing another multivariate model by including all significant factors identified in the univariate analysis.

Thank you for your note. In fact, we did include univariate analysis alongside multivariable analysis in order to reduce the number of tables. However, we did acknowledge your recommendation, and the main text was amended by inclusion of a newly designed table (Table 5) that was reserved to Multivariable analysis, while Table 4 now contains only univariate analysis.
